

# Plastome structure and adaptive evolution of *Calanthe* s.l. species

Yanqiong Chen, Hui Zhong, Yating Zhu, Yuanzhen Huang, Shasha Wu, Zhongjian Liu, Siren Lan and Junwen Zhai

Key Laboratory of National Forestry and Grassland Administration for Orchid Conservation and Utilization at College of Landscape Architecture, Fujian Agriculture and Forestry University, Fuzhou, Fujian, China
Fujian Ornamental Plant Germplasm Resources Innovation & Engineering Application Research Center, Fujian Agriculture and Forestry University, Fuzhou, Fujian, China

Corresponding authors
Siren Lan, lkzx@fafu.edu.cn
Junwen Zhai, zhai-jw@163.com

## ABSTRACT

*Calanthe* s.l. is the most diverse group in the tribe Collabieae (Orchidaceae), which are pantropical in distribution. Illumina sequencing followed by *de novo* assembly was used in this study, and the plastid genetic information of *Calanthe* s.l. was used to investigate the adaptive evolution of this taxon. Herein, the complete plastome of five *Calanthe* s.l. species (*Calanthe davidii*, *Styloglossum lyroglossa*, *Preptanthe rubens*, *Cephalantheropsis obcordata*, and *Phaius tankervilliae*) were determined, and the two other published plastome sequences of *Calanthe* s.l. were added for comparative analyses to examine the evolutionary pattern of the plastome in the alliance. The seven plastomes ranged from 150,181 bp (*C. delavayi*) to 159,014 bp (*C. davidii*) in length and were all mapped as circular structures. Except for the three *ndh* genes (*ndhC*, *ndhF*, and *ndhK*) lost in *C. delavayi*, the remaining six species contain identical gene orders and numbers (115 gene). Nucleotide diversity was detected across the plastomes, and we screened 14 mutational hotspot regions, including 12 non-coding regions and two gene regions. For the adaptive evolution investigation, three species showed positive selected genes compared with others, *C. obcordata* (*cemA*), *S. lyroglossa* (*infA, ycf1* and *ycf2*) and *C. delavayi* (*nad6* and *ndhB*). Six genes were under site-specific positive selection in *Calanthe* s.l., namely, *accD*, *ndhB*, *ndhD*, *rpoC2*, *ycf1*, and *ycf2*, most of which are involved in photosynthesis. These results, including the new plastomes, provide resources for the comparative plastome, breeding, and plastid genetic engineering of orchids and flowering plants.

## INTRODUCTION

Orchidaceae is the largest family in angiosperms (ca. 736 genera, ca. 28,000 species; *Chase et al., 2015*; *Christenhusz & Byng, 2016*), with its fascinating biodiversity attracting the interest of numerous botanists. The first plastome of orchids (*Phalaenopsis aphrodite*) was published in 2006 (*Chang et al., 2006*). In total, over 200 plastomes of orchids have been published, including five subfamilies and 51 genera. In general, plastomes are relatively conserved in land plants, but huge divergence was found among different orchid species. Orchids encompass all life forms of plants, including both heterotrophic and autotrophic

species, namely, terrestrial, epiphytic, and saprophytic (also called mycoheterotrophic). The evolutionary direction of the life forms of orchids can be roughly outlined from terrestrial to epiphytic, while saprophytic species independently evolved several times (*Sosa et al., 2016*). Recent plastome analysis of orchid species has focused on comparisons with partially and fully mycoheterotrophic species (*Feng et al., 2016*; *Barrett & Kennedy, 2018*; *Unruh et al., 2018*; *Yuan et al., 2018*). The molecular information on these species is of interest not only for the study of phylogenetics but also for evolutionary studies and species conservation.

*Calanthe*, the largest genus of the Collabieae tribe (Epidendroideae, Orchidaceae), has a pantropical distribution, being widely distributed in tropical and subtropical Asia, Australia, Madagascar, Africa, Central and South America, and the Caribbean (*Chen et al., 2009*; *Clayton & Cribb, 2013*). Since the establishment of *Calanthe* in 1821 (*Ker Gawler, 1821*), the genus has undergone multiple intra-generic taxonomic revisions lasting for centuries. For example, *Lindley (1855)* set two subgenera according to the spur length, *Bentham (1881)* and *Pridgeon et al. (2005)* defined the genus with different criteria based on the pseudobulb and floral characteristics. For the taxonomic classification, morphological data are limited. Genetic information suggests that the origin of *Calanthe* is polyphyletic and includes its relatives (*Cephalantheropsis* and *Phaius*) (*Yukawa & Ishida, 2008*; *Xiang et al., 2014*; *Zhai et al., 2014*). The redefinition of the three genera seems inevitable, and *Calanthe* may be subdivided into three genera, *Calanthe*, *Styloglossum*, and *Preptanthe* (*Yukawa & Cribb, 2014*; *Zhai et al., 2014*). These genera, *Calanthe*, *Phaius*, *Styloglossum*, *Cephalantheropsis* and *Preptanthe*, form an independent alliance (which we refer to as *Calanthe* s.l.) within Epidendroideae (Orchidaceae). This alliance can be easily distinguished from other taxa in this family, characterized by plicate leaves, similar sepals and petals, and eight waxy pollinia. However, the latest Orchidaceae classification retained the independent genus status of polyphyly *Calanthe* (*Chase et al., 2015*). To better understand their phylogenetic relationship, it is necessary for us to identify discrepancies in the genetic information of the major clade of *Calanthe* s.l.

In the current study, we assembled and annotated the plastomes of five *Calanthe* s.l. species for the first time. Additionally, we published and compared *Calanthe* s.l. plastome with the aim to: (1) understand the genetic variation within the *Calanthe* s.l. plastome; (2) identify the characteristics of the plastomes structure, sequence divergence, mutational hotspot regions, repeat regions, and examine them as candidate molecular markers for species classification and further species evolution studies; (3) assess the selective pressure among *Calanthe* s.l. species by identifying genes underlying positive selection; and (4) evaluate the phylogenetic relationships within *Calanthe* s.l.

## MATERIALS & METHODS

### Plant materials, DNA extraction and sequencing

Specimens of five species, *Calanthe davidii*, *Styloglossum lyroglossa*, *Preptanthe rubens*, *Phaius tankervilliae*, and *Cephalantheropsis obcordate*, represent the five major clades of the *Calanthe* s.l. (according to the phylogenetic relationship detected by *Zhai et al. (2014)*)

and they all introduced and cultivated in the Fujian Agriculture and Forestry University, Fujian province, China. Their voucher information is given in Table S1. The modified version of the CTAB method (*Doyle & Doyle, 1987*) was applied to extract the genomic DNA. We constructed the short-insert (500 bp) pair-end (PE) library and the sequencing was conducted by the Beijing Genomics Institute (Shenzhen, China) on the Illumina HiSeq 2500 platform, with a read length of 150 bp. At least 5 Gb clean data were obtained for each species.

## Genome assembling and annotation

We used the GetOrganelle pipe-line (*Jin et al., 2019*) to obtain plastid-like reads, with 18 orchid species as references (listed in Table S2). Then, the filtered reads were assembled using SPAdes version 3.1.0 (*Bankevich et al., 2012*). The final plastid-like reads were filtered by the script of GetOrganelle to obtain pure plastid contigs, and the filtered De Brujin graphs were viewed and edited using Bandage (*Wick et al., 2015*) to manually finalize the complete target genomes. DOGMA (*Wyman, Jansen & Boore, 2004*) was used to annotate the plastome, a web-based package based on BLAST searches against a custom database that can identify various genes. Then, we used Geneious prime v2019.03 (*Kearse et al., 2012*) to manually align the results, and we examined and confirmed the plastomes with *C. triplicata* as the reference. OrganellarGenomeDRAW (OGDRAW) version 1.3.1 (https://chlorobox.mpimp-golm.mpg.de/OGDraw.html) (*Greiner et al., 2019*) was used to visualize the structural features of the five species.

## Genome comparison and analysis

Combined with two published *Calanthe* plastome (*C. triplicata* KF753635 and *C. delavayi*, MK388860), seven complete plastomes of *Calanthe* s.l. provided the possibility of comparative analysis within the relatives. In addition, We also compared the two plastome-assembled versions of *C. davidii* (D1, MG708353) and *C. davidii* (D2, assembled in the present study). Mauve alignment (*Darling et al., 2004*) was employed to analyse the plastome DNA rearrangement of seven species. The junction regions between the IR, SSC, and LSC of seven species were compared using the online program IRscope (https://irscope.shinyapps.io/irapp/) (*Amiryousefi, Hyvönen & Poczai, 2018*). Meanwhile, the sequence identity of the seven species were compared and plotted using the program mVISTA (http://genome.lbl.gov/vista/mvista/submit.shtml) with Shuffle-LAGAN (*Brudno et al., 2003*; *Frazer et al., 2004*) with *C. triplicata* (KF753635) as reference to show inter- and intraspecific variations.

To identify the mutational hotspot regions and genes, we calculated nucleotide diversity ($\pi$) across the whole plastome. 76 CDS and 56 IGS shared by seven species were used for analysing and were extracted by PhyloSuit (*Zhang et al., 2020*). Seven plastome sequences were aligned using MAFFT v7.407 (*Katoh & Standley, 2013*) and the nucleotide diversity was detected using DnaSP v6.12.03 (DNA Sequences Polymorphism) (*Rozas et al., 2017*) with sliding window strategy. The step size was set to 200 bp, with a 600 bp window length.

## Repeat sequence analysis

The online software REPuter (https://bibiserv.cebitec.uni-bielefeld.de/reputer/) was used to identify the repeat sequences (*Kurtz et al., 2001*), including palindromic direct and reverse repeats. The parameters were set as follows: (1) The maximum and minimum computed repeat sizes were limited to 50 and 30, respectively; (2) Hamming distance of 3. Tandem repeat sequences were identified with the tandem repeats finder (*Benson, 1999*). The alignment parameters were set as 2 match, 7 mismatch, and 7 indels. We identified repeats with condition of 80 minimum alignment score, maximum period sizes in 500 bp, and maximum TR array sizes of 2 million. Perl script MISA (MIcroSAtellite identification tool) was used to detect simple sequence repeats (SSRs) loci of the plastome, with a threshold of mono-, di-, tri-, tetra-, penta-, and hexa-nucleotide SSRs, respectively (*Thiel et al., 2003*).

## Gene selection pressure analysis

To identify the gene divergences changes within seven species, nonsynonymous (Ka) and synonymous substitution rates (Ks), and the ratio Ka/Ks were calculated using the KaKs_calcaulator (*Wang et al., 2010*). We analysed all CDS gene regions (76 genes), except for *ndhC*, *ndhF* and *ndhK* as these regions were lost in *C. delavayi*, and *ycf15* as pseudogenes in *S. lyroglossa* and *P. rubens*. The seven species pairwise comparison generated 21 species pairs in total. The parameters were set as genetic code tab11 (bacterial and plant plastid code), method of calculation YN. When the genes had no substitutions on the alignment or there was 100% match, the Ks value was set as 0; in this case, the Ka/Ks value was shown as "NA" in the results. We replaced all "NA" results with 0.

We also evaluated the role of site-specific selection in 76 genes of seven species. The maximum likelihood tree was constructed based on the aligned concatenated CDS genes data set of the seven species using IQ-tree (*Nguyen et al., 2015*). We calculated the nonsynonymous (dN) substitution, synonymous (dS) substitution rates, and the dN/dS ratio (ω) in order to estimate the selection pressure with site-specific model (the option of the analyses was set to seqtype = 1, model = 0, and Nssites = 0, 1, 2, 3, 7, and 8) with CodeML program in PAML 4.9 (*Yang & Nielsen, 2002*; *Yang, 2007*). The likelihood ratio tests (LRTs) *P*-values of under three pairs of site models were calculated to detect positive selection ($p < 0.05$), including: M0 (one-ratio) vs. M3 (discrete), M1 (nearly neutral) vs. M2 (positive selection), and M7 (β) vs. M8 (β and ω).

## Phylogenetic analysis

Phylogenetic analysis was conducted using plastome sequences of the seven *Calanthe* s.l. taxa mentioned above, with *Bletilla striata* (GenBank accession No. KT588924) used as an outgroup. We generated four data sets for phylogenetic inference: (1) complete plastome sequences; (2) coding regions (CDS); (3) intergenic regions (IGS); and (4) 13 highly variable regions screened in the present study (*nad6* gene was lost in *Bletilla striata*, so *nad6-ndhI* was excluded in the analysis). We extracted the 76 CDS and 46 IGS shared by eight species from plastomes using PhyloSuit (*Zhang et al., 2020*). MAFFT v7.107 (*Katoh & Standley, 2013*) was conducted for sequence alignments.

**Table 1  The basic characteristics of the plastome of eight *Calanthe* s.l. species.**

| Species | *Calanthe triplicata* | *Calanthe davidii* (D1) | *Calanthe davidii* (D2) | *Calanthe delavayi* | *Phaius tankervilliae* | *Cephalantheropsis obcordata* | *Styloglossum lyroglossa* | *Preptanthe rubens* |
|---|---|---|---|---|---|---|---|---|
| Accession number | KF753635 | MG925365 | MN708353 | MK388860 | MN708349 | MN708351 | MN708350 | MN708352 |
| Genome size(bp) | 158,759 | 153,629 | 159,014 | 150,181 | 158,229 | 157,919 | 156,036 | 158,215 |
| LSC length(bp) | 87,305 | 86,045 | 87,857 | 83,411 | 86,638 | 86,650 | 85,421 | 87,498 |
| SSC length(bp) | 18,476 | 15,672 | 18,589 | 16,338 | 18,357 | 18,420 | 18,149 | 18,397 |
| IR length(bp) | 26,489 | 25,956 | 26,284 | 25,216 | 26,617 | 26,424 | 26,233 | 26,160 |
| Coding(bp) | 79,578 | 72,495 | 79,572 | 73,731 | 79,671 | 79,609 | 79,109 | 79,208 |
| Non-Coding(bp) | 79,181 | 81,134 | 79,442 | 76,450 | 78,558 | 78,310 | 76,927 | 79,007 |
| Number of genes | 136(115) | 136(115) | 136(115) | 133(112) | 136(115) | 136(115) | 136(115) | 136(115) |
| Number of protein-coding genes | 88(80) | 88(80) | 88(80) | 85(77) | 88(79) | 88(80) | 86(79) | 86(79) |
| Number of tRNA genes | 38(30) | 38(30) | 38(30) | 38(30) | 38(30) | 38(30) | 38(30) | 38(30) |
| Number of rRNA genes | 8(4) | 8(4) | 8(4) | 8(4) | 8(4) | 8(4) | 8(4) | 8(4) |
| GC content (%) | 36.70 | 36.90 | 36.60 | 36.90 | 37.00 | 36.80 | 36.90 | 36.70 |
| GC content in LSC (%) | 34.40 | 34.50 | 34.40 | 34.50 | 34.80 | 34.50 | 34.60 | 34.30 |
| GC content in SSC (%) | 29.70 | 30.20 | 29.60 | 29.40 | 29.90 | 29.70 | 29.90 | 29.70 |
| GC content in IR (%) | 43.00 | 43.10 | 43.10 | 43.30 | 43.00 | 43.10 | 43.10 | 43.30 |

**Table 2  Genes length (bp) difference between two versions of *Calanthe davidii* complete plastome.**

| Version | *nad6* | *ndhA* | *ndhC* | *ndhD* | *ndhE* | *ndhF* | *ndhI* | *ndhK* | *ycf2* |
|---|---|---|---|---|---|---|---|---|---|
| D1 (MG925365) | 120 | 869 | 171 | 987 | 306 | 1,827 | 297 | 138 | 4,650 |
| D2 (MN708353) | 531 | 2,215 | 363 | 1,509 | 321 | 2,259 | 504 | 774 | 6,813 |

A maximum likelihood (ML) tree was constructed using IQ-tree with 1,000 bootstrap replicates. The optimal nucleotide substitution model was found with ModelFinder module in IQ-tree (*Nguyen et al., 2015*). Bayesian inference (BI) was performed with Mrbayes v3.2 (*Ronquist et al., 2012*). The Markov chain Monte Carlo (MCMC) analysis was run for 50,000,000 generations. The stationarity was regarded as having been reached when the average standard deviation of split frequencies remained below 0.01. Trees were sampled at every 1,000 generations with the first 25% discarded as burn-in. The remaining trees were used to build a 50% majority-rule consensus tree.

## RESULTS

### Features of the plastome

We reassembled and annotated the plastome of *C. davidii* (we use the abbreviations D1 (MG925365) and D2 (MN708353) to represent the different versions of the *C. davidii* plastome). In the present study, the reassembled version of *C. davidii* (D2) is hugely different from the published plastome. A comparison between these two genomes indicated that the gene numbers and orders are identical, while huge differences were found in the genome size, gene length, and GC content (Tables 1 and 2). We used the D2 version of *C. davidii* for the subsequent analysis.

The genome size of D1 is 5,385 bp shorter than D2, with the difference mainly concentrated in the IR region, especially *ndh* genes, which encode the subunits of the nicotinamide adenine dinucleotide (NADH) dehydrogenase-like complex proteins (*Yamori & Shikanai, 2016*). The assembling and annotating of D1 used *Dendrobium nobile* as the reference (*Dong et al., 2018*), which has a discrepancy with *C. triplicata* we used as reference in this study, especially *ndh*, e.g., the *ndhC*, *ndhI*, *ndhK*, and *nad6* genes were lost in *D. nobile* but exist in *C. triplicata*. Moreover, *D. nobile* showed a distant relationship with *C. davidii*, *Dendrobium* belong to Malaxideae tribe were far away from the Collabieae tribe, to which *C. davidii* belongs (*Chase et al., 2015*). Considering the conservatism of the plastome, and the reasons mentioned above, we may infer that the results of this study are more accurate.

The plastome of the seven species of *Calanthe* s.l. ranged from 150,181 bp (*C. delavayi*) to 159,014 bp (*C. davidii,* D2), with a GC content of 36.60%–37.00% (Table 1). They all shared the common structure: a pair of IRs (IRa and IRb) (25,216–26,617 bp), separated by LSC region (83,411–87,857 bp) and SSC region (16,338–18,589 bp) (Table 1 and Fig. 1). The gene numbers, orders, and names were very similar among *Calanthe* s.l., except for the three gene losses of *C. delavayi*, namely, *ndhC*, *ndhF*, and *ndhK* (Figs. 2–4).

The genomes encoded 133–136 genes, of which 112–115 were unique genes, containing 77–80 protein-coding genes, 30 tRNA genes, and four rRNA genes. Two pseudogenes were found in the seven plastomes; the *ycf15* was the only pseudogenization in *S. lyroglossa* and *P. rubens* while the *ycf68* gene was a pseudogene in all seven species. They both contain many internal stop codons. The 18 genes were found duplicated in the IR regions, including three type of genes, namely, coding genes, tRNA genes and rRNA genes (Table 3). In the seven plastomes, 15 genes contained one intron (six tRNA and nine protein-coding genes) and three genes (*ycf3*, *clpP*, and *rps12*) contained two introns. The gene *rps12* was trans-spliced, crossing the two areas with the 5′-end exon lie in the LSC region and the intron, 3′-end exon located in the IR region. Overlapping sequences were found in three pair of genes: *trnK*-UUU/*matK*, *atpE/atpB*, and *psbD/psbC*.

## Comparative genomic analysis

The seven plastome sequence identities were plotted using mVISTA and with *C. triplicata* (KF753635) as reference. The alignment revealed high sequence similarity across the seven plastomes and no rearrangement occurred (Figs. 2 and 3) which suggested that they are highly conserved. The SSC region has the highest divergence ($\pi = 0.0187$) while IR region is the most conservative ($\pi = 0.0036$) (Table S3). We chose the nine mutational hotspots according to the sequence divergence analysis (nucleotide diversity ($\pi > 0.03$)), including 12 regions (*atpB-rbcL ccsA-ndhD*, *clpP-psbB*, *nad6-ndhI*, *psbA-matK*, *psbB-psbT*, *rpl32-trnL*-UAG, *rps11-rpl36-infA*, *rps16-trnQ*-UUG, *trnE*-UUC-*trnM*-CAU, *trnF*-GAA-*ndhJ*, *trnS*-UGA-*trnG*-GC*)*, and two gene regions (*trnK*-UUU, *clpP*) (Figs. 3 and 4).

We examined the expansion and contraction of the IR area between the single-copy regions and the pair of IR regions for the seven *Calanthe* s.l. species (Fig. 5). Among the seven species, the gene positions of four borders (LSC/IRb, IRb/SSC, SSC/IRa, and IRa/LSC) had different types. The LSC/IRb border had three situations. First, in *C. triplicata*,

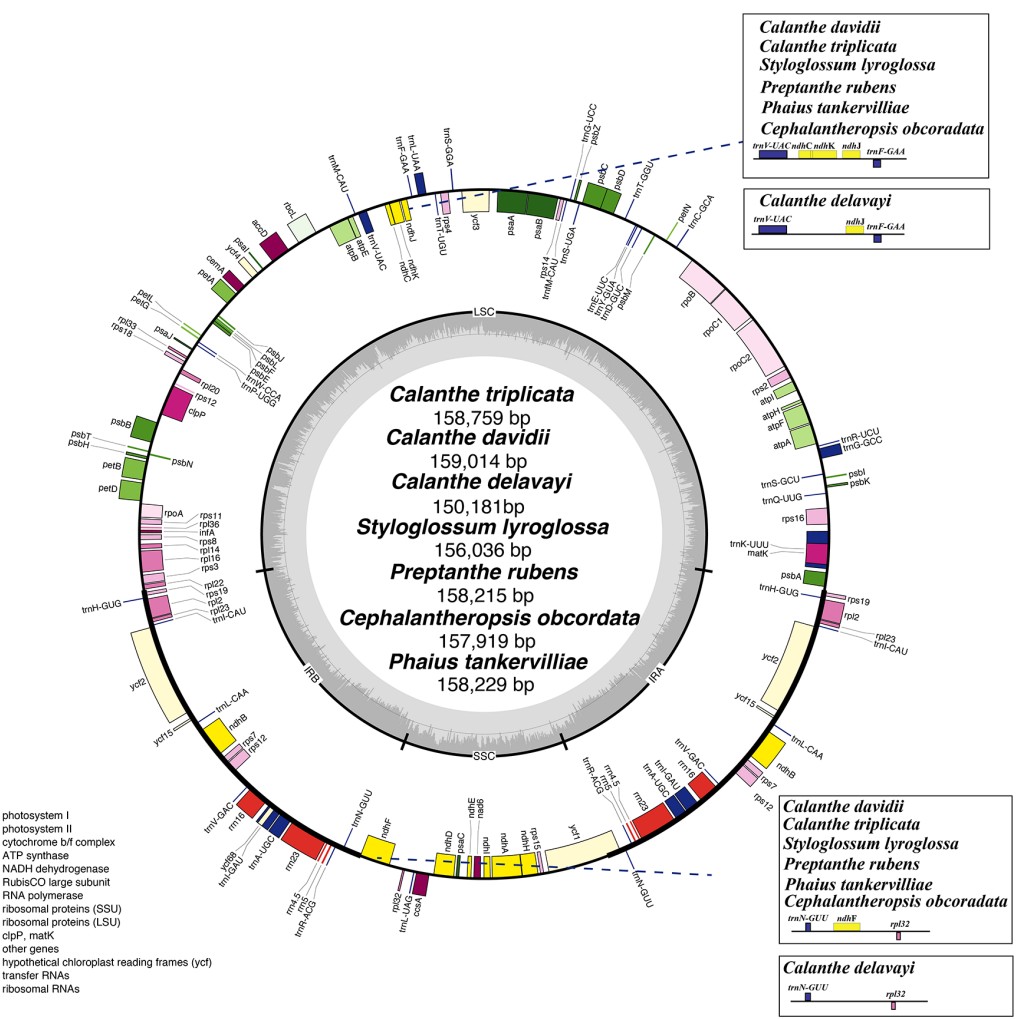

**Figure 1 Plastome map of *Calanthe* s.l.** Genes inside the circle are transcribed clockwise, and those outside are transcribed counterclockwise. Genes of different functions are color-coded. The darker gray in the inner circle shows the GC content, while the lighter gray shows the AT content.

*C. delavayi*, *P. tankervilliae*, and *C. obcordata*, the *rpl22* gene overlapped in the LSC/IRb region; second, in *C. davidii* and *S. lyroglossa* the *rpl22* gene was located in the LSC region, 9–47 bp away from the IRb region; the third situation is in *P. rubens*, where the *rps19* gene overlapped the border instead of the *rpl22* gene. While the IRb/SSC junction regions were relatively stable in the seven species, the *ndhF* gene crossed the border of six of the seven species, except for *C. delavayi*, due to its *ndhF* gene loss, and the nearest gene (*trnN* (in IRb)) is 361 bp away from the SSC region. SSC/IRa and IRa/LSC are both very conserved among the seven plastomes. The *ycf1* gene strode the SSC/IRa bounder, having 35–1,042 bp into the IRa region. The distance between *psbA* and the IRa/LSC junction ranged from 91 to 239 bp.

Repeats in the plastome were detected. *P. rubens* had the greatest number of long repeats and *C. triplicata* had the greatest number of tandem repeat regions (Tables S4 and

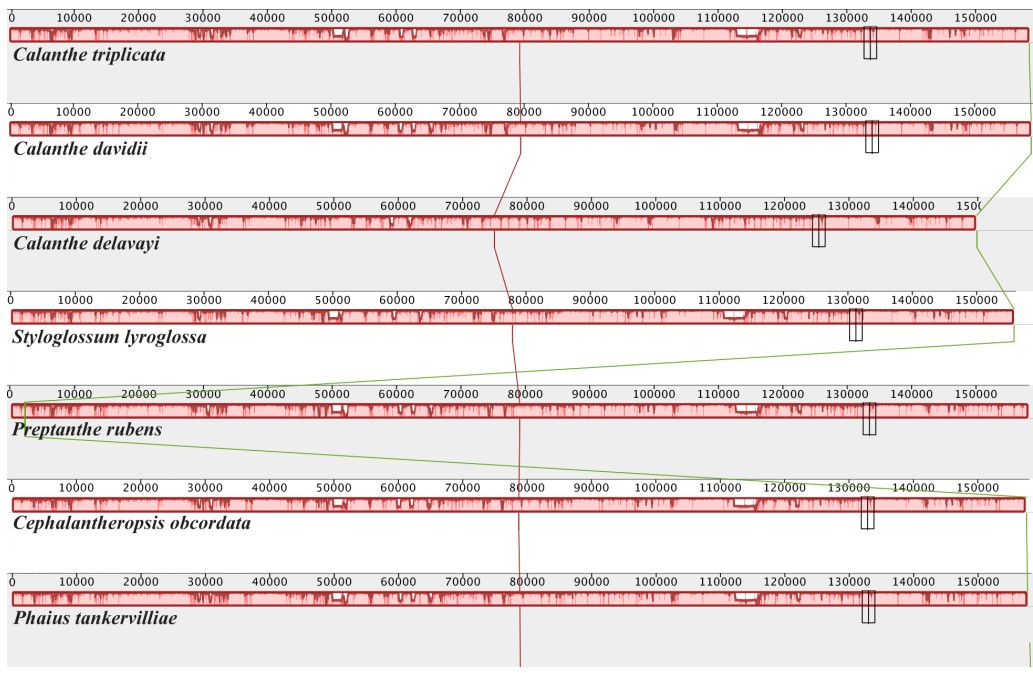

**Figure 2** **Mauve (Multiple Alignment of Conserved Genomic Sequence With Rearrangements) align-ment of platome of seven species of *Calanthe* s.l.** The *Calanthe triplicata* genome is shown at the top as the reference genome.

S5). A total number of 49–73 SSRs were found in the seven plastomes. Mononucleotide, dinucleotide, trinucleotide, tetranucleotide, and pentanucleotide SSRs were all discovered in seven *Calanthe* s.l. species (Fig. 6). Hexanucleotide SSRs were found except in *C. davidii* and *P. rubens*. In all seven species, mononucleotide repetitions accounted for more than half of all (57.89%, 56.67%, 54.79%, 61.22%, 54.68%, 53.73%, and 60%). The IGS region contained the largest number of SSRs (with 231 identified), while 175 were identified in the CDS and 64 in the coding sequence introns. In particular, all mononucleotide SSRs belonged to A or T types, and the richness of A or T were found in the major of dinucleotide, trinucleotide, tetranucleotide, pentanucleotide, and hexanucleotide SSRs (Table S6).

## Gene selection pressure analysis of protein sequences

The average Ka/Ks ratio for 76 protein-coding genes analysed in the seven genomes was 0.2455. We found the 13 most conserved genes with average Ka/Ks values between 0 and 0.01, *petG*, *psaC*, *psbE*, *psbI*, *psbJ*, *psbl*, *psbN*, *psbT*, *rpl23*, *rpl36*, *rps12*, *rps19* and *rps7*. These genes are under very strong purifying selection pressure. Values of Ka/Ks in the range of 0.5 to 1.0 (indicating relaxed selection) were observed for the genes *matK*, *atpF*, *rpoC2*, *accD*, *rps18*, *rpoA*, *ndhD*, *ndhI* and *rps15*. We found six protein-coding genes with Ka/Ks >1, including *cemA* (1.3680) in *C. obcordata*, *infA* (2.9934), *ycf1* (1.2618) and *ycf2* (1.0886) in *S. lyroglossa*, *nad6* (3.0761) and *ndhB* (1.4169) in *C. delavayi* (Table S7, and Fig. 7).

The site-specific selective pressure on seven species of *Calanthe* s.l. were assessed using the site model in the PAML program. Three pairs of site model comparisons (M0 vs. M3,

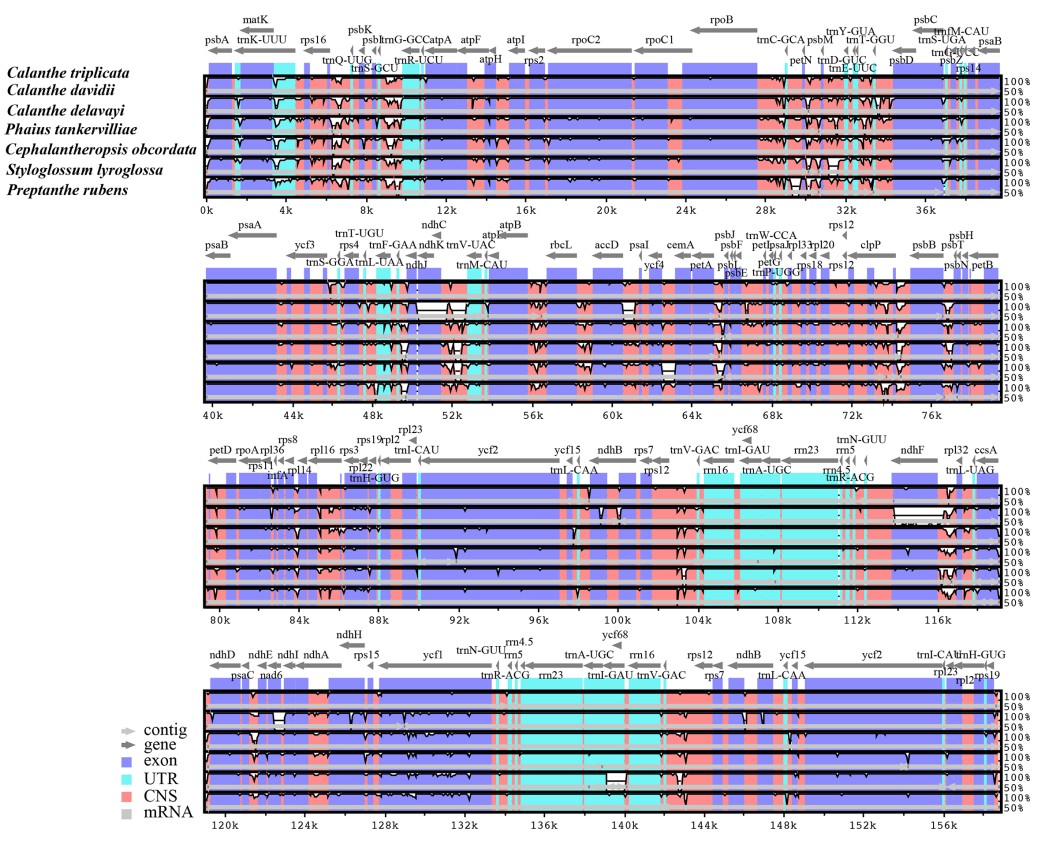

**Figure 3** **Comparison of seven *Calanthe* s.l. plastomes using mVISTA program, taking the annotation of *Calanthe triplicata* as a reference.** The top line shows the genes in order. A cut-off of 70% identity was used for the plots and the Y-scale represents the percent identity between 50 and 100%. Genome regions are color-coded as exon and conserved non-coding sequences (CNS).

M1 vs. M2a, M7 vs. M8) showed that six genes were related to two photosynthetic electron transport (*ndhD* and *rbcL*), gene expression (*rpoc2*), and other functions (*accD, ycf1* and *ycf2*) had been subjected to positive selection (LRT of the three comparison all $P < 0.05$) (Tables S8 and S9).

## Phylogenetic analysis

The phylogenetic relationship inferred by ML and BI analysis of the four data sets resulted in the same topology (Fig. 8). The seven *Calanthe* s.l. species were classified into three major clades, with all the *Calanthe* s.l. species composing a monophyly. *Preptanthe* is located at the basal position and is sister to the clade formed by the remaining other four genera. *Cephalantheropsis* and *Styloglossum* are clustered into a sister clade to *Phaius*, which forms the sister group to *Calanthe*.

 

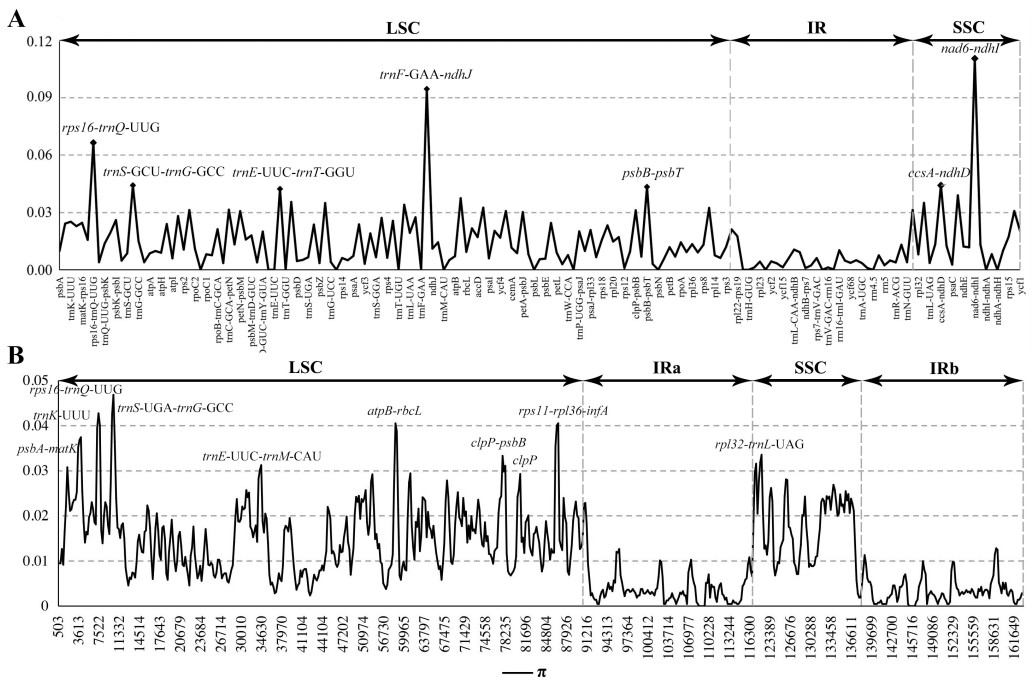

**Figure 4** **Comparison of nucleotide diversity (π) values among the seven *Calanthe* s.l.** (A) The protein coding sequences (CDS) and intergenic spacers (IGS); (B) Sliding window analysis of the whole plastome of seven species.

## DISCUSSION

### Chloroplast sequence variation

The whole plastome of five *Calanthe* s.l. species were determined, which had never been done for *Phaius*, *Cephalantheropsis*, *Styloglossum* and *Preptanthe* and was a reconstruction for *C. davidii*. Comparative analyses of the plastome of seven *Calanthe* s.l. species showed highly conserved structures and genes. The published Orchidaceae plastome genome size ranges from 35,304 bp (*Gastrodia elata*) to 178,131 bp (*Cypripedium formosanum*). This discrepancy in the plastome genome size can be explained by the different life forms, as the heterotrophic plants that do not photosynthesise have lost the photosynthesis-related genes. In the present study, only *C. delavayi* lost the *ndh* genes (*ndhC*, *F*, and *K*) in *Calanthe* s.l. It is also common to see a lack of functional *ndh* genes in other autotrophic orchids, i.e., *Cymbidium*, *Dendrobium*, *Phalaenopsis* and *Ophrys* (*Yang et al., 2013*; *Lin et al., 2017*; *Roma et al., 2018*). Although the transfer of the *ndh* genes from the plastome genome to the mitochondrial (mt) genome was detected in some orchids, there is no direct evidence showing that these transfers were linked to the losses of *ndh* in the plastome (*Lin et al., 2015*). The mechanisms for the complex deletion and truncation of the genes encoding NADH dehydrogenase subunits in orchids remain unclear.

We found two pseudogenes of the seven species, two hypothetical genes *ycf15* and *ycf68*. The *ycf15* was first identified as ORF87 in the *Nicotiana* chloroplast genome (*Shinozaki et al., 1986*); however, the validity of the gene as protein-coding gene has been questioned

**Table 3  Gene contents in seven *Calanthe* s.l. species plastomes.**

| Classfication | Genes |
|---|---|
| Genetic apparatus | |
| Large ribosomal subunits | *rpl2*[a]( ×2), *rpl14*, *rpl16* [a], *rpl20*, *rpl22*, *rpl23* ( ×2), *rpl32*, *rpl33*, *rpl36* |
| Small ribosomal subunits | *rps2*, *rps3*, *rps4*, *rps7* ( ×2), *rps8*, *rps11*, *rps12*[b], *rps14*, *rps15*, *rps16*[a], *rps18*, *rps19* (×2) |
| RNA polymerase subunits | *rpoA*, *rpoB*, *rpoC1*[a], *rpoC2* |
| DNA dependent RNA polymerase Protease | *clpP*[b] |
| Maturase | *matK* |
| Ribosomal RNAs | *rrn4.5*(×2), *rrn5* (×2), *rrn23* (×2), *rrn16* (×2) |
| Transfer RNAs | *trnA*-UGC(×2)[a], *trnC*-GCA, *trnD*-GUC, *trnE*-UUC, *trnF*-GAA, *trnfM*-CAU, *trnG*-GCC, *trnG*-UCC[a], *trnH*-GUG(×2), *trnI*-CAU(×2), *trnH*-GUG, *trnI*-GAU(×2)[a], *trnK*-UUU[a], *trnL*-CAA(×2), *trnL*-UAG, *trnL*-UUA[a], *trnM*-CAU, *trnN*-GUU(×2), *trnP*-UGG, *trnQ*-UUG, *trnR*-ACG(×2), *trnR*-UCU, *trnS*-GCU, *trnS*-GGA, *trnS*-UGA, *trnT*-GGU, *trnV*-GAC(×2), *trnV*-UAC[a], *trnW*-CCA, *trnY*-GUA |
| Light dependent photosynthesis | |
| Photosystem I | *psaA*, *psaB*, *psaC*, *psaI*, *psaJ*, *ycf3* [b], *ycf4* |
| Photosystem II | *psbA*, *psbB*, *psbC*, *psbD*, *psbE*, *psbF*, *psbH*, *psbI*, *psbJ*, *psbK*, *psbL*, *psbM*, *psbN*, *psbT*, *psbZ* |
| NAD(P)H dehydrogenase complex | *ndhA*[a], *ndhB*[a](×2), *ndhC*[c], *ndhD*, *ndhE*, *ndhF*[c], *ndhG*, *ndhH*, *ndhI*, *ndhJ*, *ndhK*[c] |
| F-type ATP synthase | *atpA*, *atpB*, *atpE*, *atpF*[a], *atpH*, *atpI* |
| Cytochrome b6/f complex | *petA*, *petB*[a], *petD* [a], *petG*, *petL*, *petN* |
| Light independent photosynthesis | |
| Inner membrane protein | *cemA* |
| Cytochrome C biogenesis protein | *ccsA* |
| Large subunit of Rubisco | *rbcL* |
| Subunit of acetyl-CoA-carboxylase | *accD* |
| Translation initiation factor | *infA* |
| Function uncertain | *ycf1*, *ycf2* (×2), *ycf15* (×2), *ycf68* (×2) |

**Notes.**
[a]Gene containing one intron.
[b]Gene containing two introns, a trans-splinting gene, (2) shows genes have two copies.
[c]Gene lost in *C. delavay*.

(*Chumley et al., 2006*). It was disabled in *Amoborella*, *Nuphar* and most rosids (*Goremykin et al., 2003*; *Raubeson et al., 2007*). In some lineages it has been completely lost, for example *Illicium*, *Acorus*, *Ceratophyllum*, *Ranunculus* and some non-photosynthetic orchids (*Shi et al., 2013*; *Yuan et al., 2018*). The likelihood that the *ycf15* and *ycf68* are not protein-coding genes has been discussed in previous research (*Raubeson et al., 2007*). In the present study, *ycf15* was annotated as a protein-coding gene in five other species (*C. triplicata*, *C. davidii*, *C. delavayi*, *C. obcordata* and *P. tankervilliae*). Whether these sequences represent protein-coding regions and their possible functions in chloroplasts needs to be further studied.

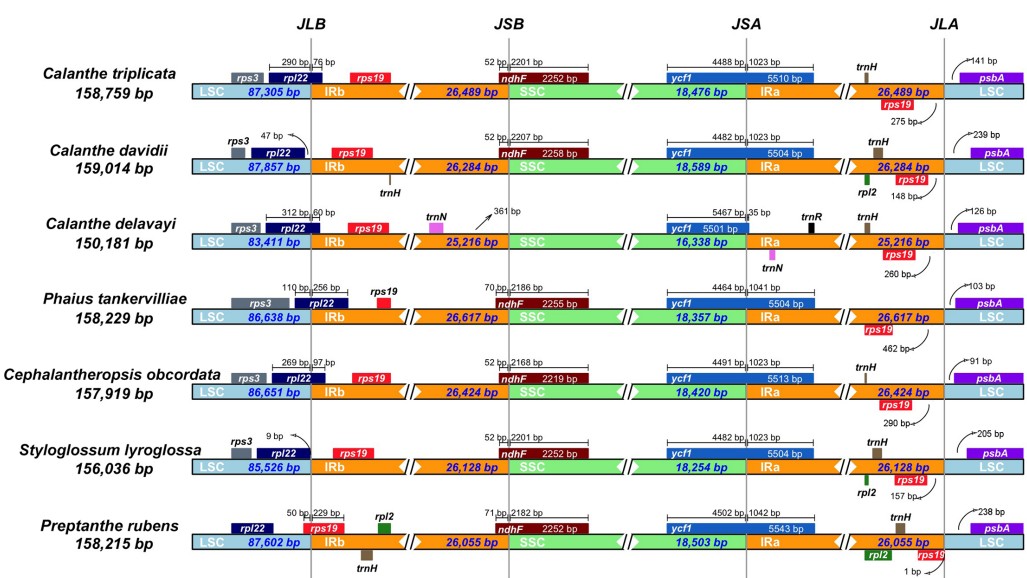

**Figure 5** **Comparison of the borders of LSC, SSC, and IR regions in seven *Calanthe* s.l. complete chloroplast genomes.** JLB (IRb /LSC), JSB (IRb/SSC), JSA (SSC/IRa) and JLA (IRa/LSC) denote the JSs between each corresponding region in the genome.

Although the overall genomic structures and gene orders of *Calanthe* s.l. are highly conserved, significant differences at the IR/SSC junction area were detected. The different expansion and contraction situation of the IR junction area will cause the plastome size differentiation (*Bock & Knoop, 2012*). In present study, among the four boundaries, only LSC/IRb shows three types in the seven species, while the remaining three (IRb/SSC, SSC/IRa, and IRa/LSC) are conservative and stable. Because of the *ndhF* gene loss in *C. delavayi*, IR contraction was detected. Previous research has pointed out that the deletion of the *ndh* genes has great influence on the instability of the IR/SSC boundary in orchids (*Kim et al., 2015*; *Niu et al., 2017*). The position of the boundary, especially the expansion and contraction of the region, could shed light on the evolution of the lineage. However, in *Calanthe* s.l., our observations would not provide the information required to elucidate the evolutionary relationships of the taxa, and whether it can benefit from adaptation require further investigation; thus, additional sampling of *Calanthe* spp. and related genera will allow for explicit tests.

## Molecular markers and candidate SSRs

Coding regions and conserved sequences of the plastome are widely used for phylogenetic inferences at higher taxonomic levels (family or genus) (*Givnish et al., 2015*; *Jansen et al., 2007*). Plastomes are ideal resources for selecting the mutational hotspots of various lineages and are used for intraspecies discrimination and phylogenetic studies at the species level (*Ahmed et al., 2013*; *Liu et al., 2018*). At present, some plastid DNA fragments are used in the taxonomy of *Calanthe* s.l., for instance *matK*, *rbcL*, and intergenic spacer *trnL-trnF* (*Zhai et al., 2014*; *Guo et al., 2017*), but they could not provide sufficient phylogeny signals

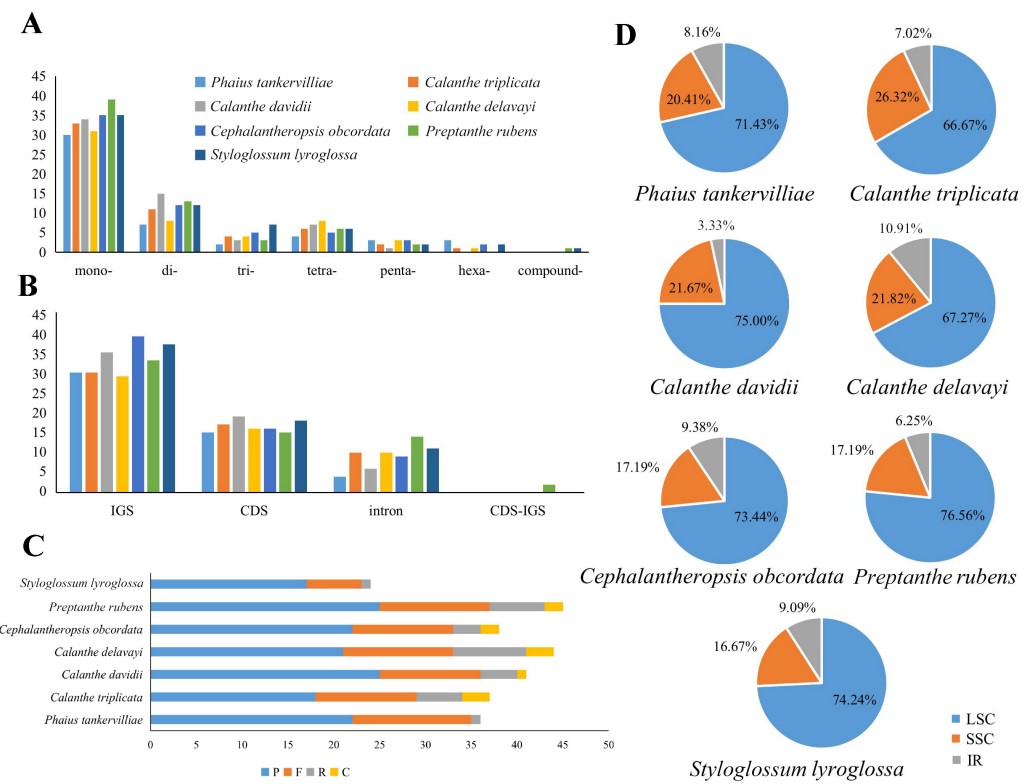

**Figure 6** Maps of repeat sequence analyses in seven *Calanthe* s.l. species plastome genome. (A) Classification of SSRs by repeat type. mono-, mononucleotides; di-, dinucleotides; tri-, trinucleotides; tetra-,tetranucleotides; penta-, pentanucleotides; and hexa-, hexanucleotides. compound-, compound formation. (B) Classification of SSRs in seven species. IGS, intergenic spacer; CDS, coding sequence, CDS-IGS, part in CDS and part in IGS. (C) Number of the four repeat types, F, P, R, and C, indicate the long repeat type (F: forward, P: palindrome, R: reverse, and C: complement, respectively). (D) SSRs locus distribution among three different regions.

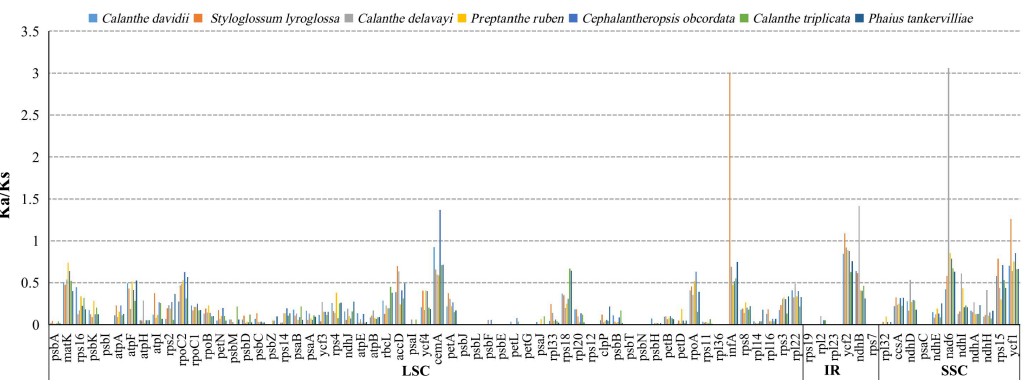

**Figure 7** The Ka/Ks ratio of 76 protein-coding genes of seven *Calanthe* s.l. plastomes.

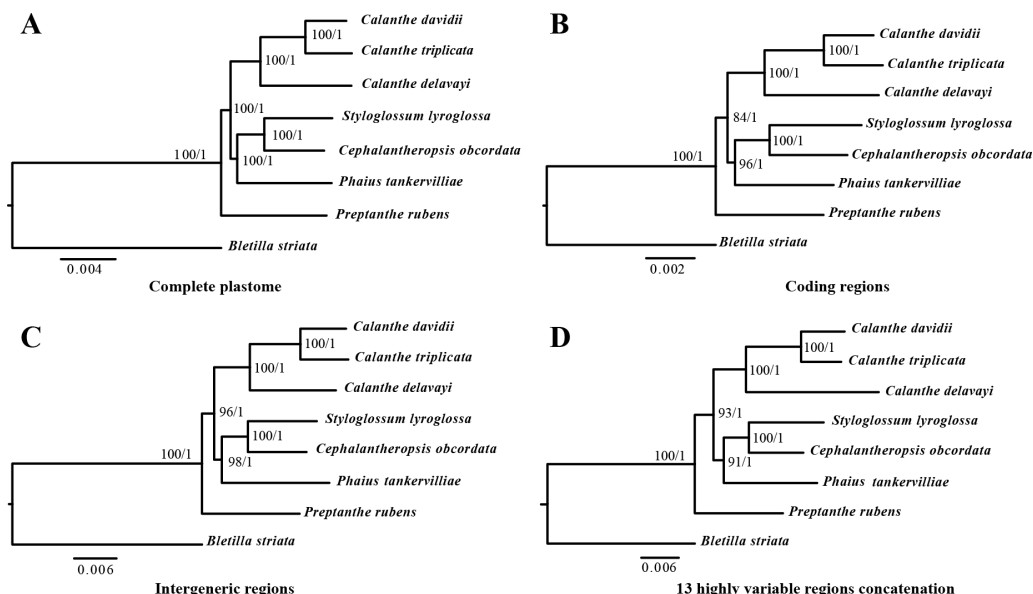

**Figure 8   Phylognetic relationships of the seven *Calanthe* s.l. taxon constructed by four DNA data sets.** Including (A) whole plastome sequences, (B) coding regions, (C) intergeneric regions, and the (D) 13 highly variable regions concatenation with Maximum likelihood (ML), and Bayesian inferencee (BI) methods. ML topology shown with ML bootstrap support value/Bayesian posterior probability listed at each node.

to establish the high-resolution phylogeny relationship for classification of related taxa, especially some infrageneric taxa whose taxonomic classification status are unclear. Our alignment screened the top 14 loci that most likely contained the highest degrees of genetic variability in *Calanthe* s.l., which can be useful in species-level phylogenetic studies of *Calanthe* s.l..

Simple sequence repeats are short (1–5 bp) repeat motifs that are tandemly repeated for varying numbers of times (*Kantety et al., 2002*). Because of its extensively dispersal in genomes, they are widely utilized in population genetics and molecular evolution studies (*Guichoux et al., 2011*). SSRs can provide interspecific polymorphisms, which are effective markers in population genetic analysis. We identified 18 SSRs as polymeric SSRs between *Calanthe* s.l. species; however, further experiments are needed to verify its effectiveness (Table 4). Mutational hotspots and SSRs derived from the plastome can serve as valuable tools for elucidating the evolutionary relationships and plant identification of *Calanthe* s.l..

## Adaptive evolution analysis

*Calanthe* s.l. are pantropical in their distribution, with high geographic and ecological diversity, from obligate epiphytic to hemi-epiphytic and terrestrial, ranging from sea level to alpine mountain areas. However, the majority live under tropical woods and forests, very often in deep shade (*Clayton & Cribb, 2013*; *Stone & Cribb, 2017*). To better understand the evolutionary history of these groups, the analysis of the genetic diversity and adaptive evolution of *Calanthe* s.l. is essential. Positive selection genes played an important part in the adaption to various environments. Within the seven species, we detected the six

**Table 4** The polymorphic simple sequence repeats in *Calanthe* s.l. plastomes.

| Type | *C. triplicata/C. davidii/C. delavayi/S. lyroglossa/P. rubens/C. obcordata/P. tankervilliae* | Location | Region |
|------|------|------|------|
| AT | 6/6/0/6/0/0/0 | *rpoB-trnC*-GCA | LSC |
| AT | 7/7/0/0/5/0/0 | *trnE*-UUC-*trnT*-GGU | LSC |
| AT | 0/0/7/6/5/7/0 | *trnL*-UAA-*trnF*-GAA | LSC |
| AT | 0/0/7/6/0/7/0 | *trnL*-UAA-*trnF*-GAA | LSC |
| GA | 5/5/5/0/5/5/5 | *ycf2* | IR |
| TA | 5/7/0/5/0/6/0 | *psbB-psbT* | LSC |
| TA | 5/5/0/0/6/0/0 | *ndhF-rpl32* | SSC |
| TA | 0/0/8/5/5/0/5 | *clpP-psbB* | LSC |
| TC | 5/5/5/0/5/5/5 | *ycf2* | IR |
| TG | 5/5/5/0/5/5/5 | *rpl33-rps18* | LSC |
| AAT | 4/4/4/4/0/4/0 | *psaC-ndhE* | SSC |
| AGAA | 0/0/3/0/3/3/0 | *psbM-trnD*-GUC | LSC |
| AATG | 0/3/3/0/3/0 | *cemA* | LSC |
| ATTA | 3/3/0/3/3/0/0 | *psaJ-rpl33* | LSC |
| GTCT | 3/3/3/3/0/3/3 | *atpA* | LSC |
| TTGA | 3/3/3/3/0/3/3 | *ndhE* | SSC |
| ATCTT | 3/3/0/0/3/3/3 | *psbK-psbI* | LSC |
| ACAAA | 3/0/0/0/0/3/3 | *ndhC-trnV*-UAC | LSC |

species-specific positive selection genes in only three species, namely, one in *C. obordata* (*cemA*), three in *S. styloglossa* (*infA*, *ycf1* and *ycf2*), and two in *C. delavayi* (*nad6* and *ndhB*). Those identified selected genes may have undergone certain functional diversification during their evolutionary history.

*cemA* encodes a chloroplast envelope membrane protein (*Sasaki et al., 1993*) and is inferred to indirectly influence $CO_2$ uptake in plastid (*Rolland et al., 1997*). In some non-photosynthetic orchids, *cemA* exists as pseudogene or is lost (*Feng et al., 2016*; *Kim et al., 2020*). *infA* is crucial for genetic information transmission that affects transcription of DNA into RNA and translation of RNA to protein; it is seen as a housekeeping gene (*Schelkunov et al., 2015*). Two genes with large ORFs encoding proteins of the unknown function, *ycf1* and *ycf2*. *nad6* gene encoding NADH-plastoquinone oxidoreductase subunit 6, *ndhB* gene encoding NAD(P)H-quinone oxidoreductase subunit 2 can influence the cyclic electron flow around photosystem I (*Shikanai et al., 1998*). They are both NAD(P)H-plastoquinone-oxidoreductase (NDH complex) encoding-related genes. Chloroplast NAD(P)H dehydrogenase is sensitive to strong light stress and can protect plants from photoinhibition or photooxidation stress and alleviate decreases in the photosynthetic rate and growth delay caused by drought, suggesting that the *ndh* genes encoding the NAD(P)H dehydrogenase (NDH) may also be involved in stress acclimation through the optimization of photosynthesis (*Horváth et al., 2000*; *Rumeau, Peltier & Cournac, 2007*). *C. delavayi* is one of species with the highest elevation distribution in the *Calanthe* s.l. (up to 3500 m). The positive selection signal of *ndhB* and *nad6* in the *C. delavayi* might be the result of adaptation to different environment with other six species.

Six genes were under positive selection in seven *Calanthe* s.l. species. Of these genes, one functions as a subunit of acetyl-CoA-carboxylase (*accD*), two NADH dehydrogenase subunit genes (*ndhB* and *ndhD*), one RNA polymerase subunit (*rpoC2*), and two genes whose functions are uncertain (*ycf1* and *ycf2*). Except for the *ndh* genes, the remaining four positively selected genes were also detected in orchid species (*Dong et al., 2018*), and these four genes may have played significant roles in the adaptive evolution history of *Calanthe* s.l. Specific roles need to be further studied.

The function of the plastid *accD* gene has been reported as essential for plant leaf development (*Kode et al., 2005*). We detected the two positively selected sites in *accD* genes for *Calanthe* s.l.. The *ndh* gene family in the plastome is involved in photosynthesis. The *ndhB* and *ndhD* genes possessed 12 and 10 positively selected sites, respectively. Although there is the viewpoint that NDH activity may not be required in some plants (*Yang et al., 2013*), these two genes may play important roles in the adaptation of *Calanthe* s.l. species to deep shade environments. In addition, the *rpoC2* (RNA polymerase subunit C2) gene was crucial for gene transcription (*Xie, Jäger & Potts, 1989*), and only one site was positively detected in our study. Plastomes contain a number of uncertain genes. The *ycfs* (hypothetical open reading frame) gene has great application potential for elucidating plant phylogeny (*Neubig et al., 2009*; *Huang, Sun & Zhang, 2010*; *Dong et al., 2015*). As the largest genes in *ycfs*, *ycf1* and *ycf2* have shown positive selection in one and eight sites, respectively, and the phenomenon extends to the many plant lineages, including orchids (*Greiner et al., 2008*; *Huang, Sun & Zhang, 2010*; *Carbonell-Caballero et al., 2015*).

## Phylogenetic analysis

Plastid sequences have been used in phylogenetic analyses based on their most nonrecombinant and uniparentally inherited and on their slower evolutionary rates than nuclear and mitochondrial genomes (*Wolfe, Li & Sharp, 1987*; *Birky, 1995*). The plastid region of *matK*, *rbcL* and *trnL-F* have been used as genetic markers with great success in Orchidaceae (*Cameron et al., 1999*; *Salazar et al., 2017*; *Hu et al., 2020*). However, the limited loci in phylogenetic inference are not powerful enough when closely related species are under consideration. Phylogenetic analyses based on plastome data sets have become a popular and practical approach and therefore comparative genomic studies of more plastome sequences have become necessary.

In the present study, the relationships among seven *Calanthe* s.l. species all had high support value (bootstrap support value > 90 and Bayesian posterior probability > 0.95), and they can be completely distinguished from one another (Fig. 8). The seven species were separated into three evolutionary branches. The branch including *C. davidii*, *C. triplicata* and *C. delavayi* was a sister to branch containing *S. lyroglossa*, *C. obcordata* and *P. tankervilliae*, and *P. rubens* was placed at the basal position. We obtained high resolution using the nine highly variable regions with lower cost.

## CONCLUSIONS

In this study, we assembled and analysed five new complete plastome sequences of *Calanthe* s.l. and compared them with other *Calanthe* species for the first time. The annotation and

comparison within *Calanthe* s.l. species showed conservative in gene sequence, GC content and genomic composition. The repeated sequences, 18 microsatellites and 14 highly mutational hotspot regions were identified in the *Calanthe* s.l. plastome. Six site-specific positively selected genes were detected. These genes will lead to understanding of the adaptations of *Calanthe* s.l. species to deep shade environments. The study will help to resolve the phylogenetic relationships and understand the adaptive evolution of *Calanthe*. It will also provide genomic resources and potential markers suitable for future species identification and speciation studies of the genus.

### Funding

This research was funded by Government Business Entrustment Project of National Forestry and Grassland Administration, grant number 2019073046 and the Key Laboratory of National Forestry and Grassland Administration for Orchid Conservation and Utilization Construction Funds, grant number 115/118990050 & 115/KJG18016A. The funders had no role in study design, data collection and analysis, decision to publish, or preparation of the manuscript.

### Grant Disclosures

The following grant information was disclosed by the authors:
National Forestry and Grassland Administration: 2019073046.
Key Laboratory of National Forestry and Grassland Administration for Orchid Conservation and Utilization Construction Funds: 115/118990050, 115/KJG18016A.

### Competing Interests

The authors declare there are no competing interests.

### Author Contributions

- Yanqiong Chen conceived and designed the experiments, performed the experiments, analyzed the data, prepared figures and/or tables, and approved the final draft.
- Hui Zhong performed the experiments, analyzed the data, prepared figures and/or tables, and approved the final draft.
- Yating Zhu and Yuanzhen Huang analyzed the data, prepared figures and/or tables, and approved the final draft.
- Shasha Wu, Zhongjian Liu, Siren Lan and Junwen Zhai conceived and designed the experiments, authored or reviewed drafts of the paper, and approved the final draft.

### Data Availability

The five chloroplast genome sequences are available at GenBank: MN708349 to MN708353.

## Supplemental Information

Supplemental information for this article can be found online at http://dx.doi.org/10.7717/peerj.10051#supplemental-information.

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
