# Peer review of "Plastome structure and adaptive evolution of Calanthe s.l. species"

_PeerJ, doi:10.7717/peerj.10051_

## Round 0.1 · original submission · Major Revisions

We have now received three reviewer reports on this manuscript. The reviewers found the general topic of the paper to be an important question in plant evolution. However, the reviewers pointed out a series of shortcomings with the study design that cast doubt on the suitability of the dataset to reach robust conclusions on the questions addressed –e.g., the number of species used to infer phylogenetic relationships between Epidendroideae is not appropriate, the identification of barcode markers is not suitable, and more details about the analyses performed should be informed. They also pointed out several misgivings about the way the manuscript has been written up. They provided very constructive comments on how the manuscript can be improved. Furthermore, I strongly recommend authors to investigate the gene divergences (Ks, Ka, Ka/Ks) to identify changes within the Calanthe species. Also, the authors need to report and discuss the presence of pseudogenes - they can be identified in Figure 5. I hope that you will find all advice helpful when revising the manuscript.

·

Basic reporting

All of these aspects are fine.

Experimental design

Experimental design is appropriate, but the reserach question is not clearly defined and strays into higher-level orchid phylogenetics, to which topic five closely plastome sequencs contributes very little. Methods etc. are all good.

Validity of the findings

Generally the findings are OK, but the higher-level analysis of Epidedroideae should be deleted, for the reasons provided below.

Additional comments

The ability to easily and quickly sequence whole plastid genomes has increased in recent years and has now made it possible to produce these for low-level taxonomic and population genetic studies. For example, Dodsworth et al. (2020) has done this for multiple accessions of Nicotiana sect. Suaveolentes, but unfortunately in some group there is pervasive retention of ancestral polymorphisms that make plastid DNA data unreliable for assessing infra-specific and low-level interspecific relationships/biogeographic patterns. Nonetheless, these data provide potentially useful tools for low-level taxonomic and population-level studies. This paper provides information about the most variable loci and polymorphic regions in Calanthe s.l., which can provide these sorts of tools for this and related genera. It is thus worthwhile publishing, but I do feel that the focus of the paper needs to be refined because it seems that the authors are more focused on higher-level orchid phylogenetics, and for this the paper is poorly suited. I recommend that the paper be refocused solely on Calanthe s.l. low-level studies and delete the orchid phylogenetic analysis, which contributes nothing to what is known about higher-level phylogenetics and sets a precedent for every future small set of orchid plastome sequences to be included in a re-analysis all previously published plastomes. This is not the way to improve what is known about higher-level relationships of Orchidaceae.

Unfortunately, the most variable genes and hypervariable non-coding regions are not generalizable in most groups of plants, including orchids. The regions found to be variable in Niu et a. (2017. The complete plastome sequences of four orchid species: insights into the evolution of Orchidaceae and the utility of plastomic mutational hotspots. Front Plant Sci. 8: 715. pmid:28515737) are not the same as in Smidt et al. (2020. Characterization of sequence variability hotspots in Cranichideae plastomes (Orchidaceae, Orchidoideae). PLoS One 15: e02278991. https:// doi.org/10.1371/journal.pone.0227991), so this paper should be focused on the value of these genomic sequences in Calanthe s.l.. The authors claim in the Abstract that “Phylogenetic analysis of Epidendroideae based on 62 plastomes indicates a close relationship between Calanthe s.l. and Eria, and the phylogenetic position of 11 tribes of Epidendroideae were clarified. They also admit that their findings are identical those portrayed in previous studies and conclude that although they “retrieved the highly resolution phylogeny of the Epidendroideae, to obtain a reliable inference a comprehensive sampling of the subfamily is necessary as limited taxon sampling can result in different tree topologies (Puslednik and Serb, 2008).” In other words, to improve our understanding of Epidendroideae relationships we should not be increasing sampling by adding only five new closely related plastomes but rather sampling more intensively across the subfamily and adding a hundred or more plastome sequences.

Other aspects of the paper seem to be conducted properly and to have provided useful results. The English is not always handled well (as in the quoted sentences above), making some sentences difficult to understand. It needs to be gone over by a native speaker to clear up these problems. I did some of this at the start of the paper, but I gave up because the job is too big.

Finally, the correct name for these genomic sequences is “plastome” or “plastid genome”, not the “chloroplast genome” or “cp genome”. The amplification of these sequences was done from whole cellular DNA, which includes chloroplasts, amyloplasts, chromoplasts etc., which all share the same genome, the plastid genome. Chloroplast genome is widely used in the literature, but this is incorrect.

·

Basic reporting

'no comment'

Experimental design

The manuscript is well written and meets the standard of experimental design in a comparative analysis of chloroplast genomes. However, I believe that the gene annotation can be better detailed and that some methods are not clear enough (such as the analysis of the diversity hotspot regions). I made some comments and suggestions line-by-line in the parts that I believe can improve.

Validity of the findings

I suggest that the authors reconsider the relationship between some SSR regions and possible barcode markers in this manuscript. A more refined analysis with more individuals would be necessary for this type of assumption.

Additional comments

Line 21 – I think that the number of chloroplast genomes can be updated;
Line 52 - References are necessary for this statement. Maybe put some information about the number of species would make that phrase more precise.
Line 107 - If there are more than 200 orchid chloroplast genomes available, why were only these 18 species chosen as references? What was the criterion for selecting these 18 species?
Line 109 - The name of the program is "Geneious Prime" and not "Geneious Primer." Geneious Primer is a suite with an extensive set of genome annotation tools. I think the information on the annotation of chloroplast genomes needs to be more detailed.
Line 120 – The program mVISTA needs a reference genome to calculate the sequence identity of each genome in the analysis. What was the reference sequence used in this analysis?
Line 122 - As the analysis performed by mVista is based on similarity with a reference genome, I believe it is not the best way to identify mutational hotspots. I think that an analysis of nucleotide diversity across the genome is the most appropriate way to identify mutational hotspots.
Line 281 - I am not sure if it is a good idea to list microsatellites as possible barcode markers in this case. Only one specimen of each species was sampled, and an experimental design that calculates the intraspecific variation of these regions is a more robust way of assessing whether these SSRs are promising for DNA Barcode. I believe that the results obtained in this article do not support this type of statement.

Reviewer 3 ·

Basic reporting

The manuscript needs proofreading. There are many typos and unclear sentences.

Experimental design

The material and methods lacks details.

Validity of the findings

The SSR regions are not promise for DNA barcode.

Additional comments

In the submitted manuscript, Chen et al. sequenced five plastid genomes from Calanthe s.l.. The authors performed a detailed characterization of these plastomes and phylogenetic inferences within Epidendroideae. They also screened the most variable regions in these plastomes. These data are of broad interest, including for future studies of population genetics and evolution in orchids.
However, the material and methods lacks details about the plastome assembly. The results on genome depth of coverage, number of reads, % of plastid reads are lacking.
The MS needs proofreading, some regions are highlighted in yellow in the pdf.
The authors should avoid unspecific expressions (highly variable, significantly shorter, relatively low, etc.).
The authors pointed out that they have chosen Calanthe because of its variation in life form, from obligate epiphytic to hemi-epiphytic and then terrestrial. However, this matter is barely discussed throughout the manuscript.
More details were provided as comments in the pdf file.

Annotated reviews are not available for download in order to protect the identity of reviewers who chose to remain anonymous.

---

## Round 0.2 · accepted · Accept

Thank you for carefully addressing all points raised by reviewers. I would gladly recommend acceptance of this revised version.

·

Basic reporting

Ok.

Experimental design

Ok.

Validity of the findings

Ok.

Additional comments

The authors did a good job of revising the manuscript.

Reviewer 3 ·

Basic reporting

No comment.

Experimental design

No comment.

Validity of the findings

No comment.

Additional comments

The manuscript has been carefully reviewed by the authors and can be accepted for publication.